# Pathological Characterization and Risk Factors of Splenic Nodular Lesions in Dogs (*Canis lupus familiaris*)

**DOI:** 10.3390/ani14050802

**Published:** 2024-03-05

**Authors:** Gloria Corvera, Raúl Alegría-Morán, Federico Francisco Cifuentes, Cristian Gabriel Torres

**Affiliations:** 1Department of Clinical Sciences, Faculty of Animal and Veterinary Sciences, Universidad de Chile, Santiago 8820808, Chile; gloria.corvera.c@gmail.com; 2School of Veterinary Medicine, Faculty of Natural Resources and Veterinary Medicine, Universidad Santo Tomás, Santiago 8370003, Chile; ralegria2@santotomas.cl; 3Laboratory of Animal Pathology ESPA, Santiago 7510612, Chile; 4Centralized Veterinary Research Lab, Faculty of Animal and Veterinary Sciences, Universidad de Chile, Santiago 8820808, Chile

**Keywords:** spleen, canine spleen tumors, hemangiosarcoma, hyperplasia

## Abstract

**Simple Summary:**

The spleen is an organ that can be affected by neoplastic and non-neoplastic lesions, which can induce diffuse, nodular, or multinodular organ growth. In dogs, few studies relate microscopical diagnosis of splenic lesions to tumor size and the number of nodules in spleen biopsies. Studying these potential associations could improve the diagnostic approach to these types of injuries, detected through several imaging techniques. This study aimed to characterize splenic masses and determine some risk factors for spleen tumors in dogs. The most frequent neoplastic and non-neoplastic diagnoses were hemangiosarcoma and hyperplasia, respectively. Most of the cases occurred in male, senior, and purebred individuals such as Cocker Spaniel, German Shepherd, and Labrador Retriever. It was determined that male sex, presence of two or more splenic nodules, and an increase in nodule size greater than 2 cm are risk factors for the presentation of splenic neoplasia, knowledge that can improve the diagnostic management of dogs with spleen lesions.

**Abstract:**

In dogs, the spleen is a secondary lymphoid organ that can be affected by both neoplastic and non-neoplastic nodules. In general, few studies relate histopathological diagnosis to tumor size and the number of nodules in spleen biopsies. Some of these studies are inconclusive regarding the difference between neoplastic and non-neoplastic lesions and have small sample sizes or do not consider all splenic lesions. This study aimed to characterize splenic masses and determine risk factors for spleen tumors in dogs. A total of 507 histological reports corresponding to the diagnosis of splenic lesions in dogs from a private laboratory of animal pathology in the Metropolitan Region, Chile, were used. Data were analyzed by descriptive statistics and multiple logistic regression. The most frequent neoplastic and non-neoplastic diagnoses were hemangiosarcoma and hyperplasia, respectively. Most of the cases occurred in male (265 cases, 52.3%), senior (421 cases, 83%), and purebred individuals (342 cases, 67.5%). The most affected breeds were the Cocker Spaniel, German Shepherd, and Labrador Retriever. The most frequent lesion was a single nodule. The variables that exhibited a greater risk for the presentation of splenic neoplasia were male sex (odds ratio (OR) = 16.21; 95% confidence interval (CI) 1.741–150.879; *p* = 0.014), the presence of two or more splenic nodules (OR = 3.94; 95% CI 2.168–7.177; *p* < 0.001), an increase in nodule size greater than 2 cm (OR for quartiles 2, 3 and 4 of 2.2; 95% CI 1.036–4.941; *p* = 0.041, 2.9; 95% CI 1.331–6.576; *p* = 0.008, and 3.6; 95% CI 1.562–8.499; *p* = 0.003, respectively), and increasing age (OR = 1.23; 95% CI 1.048–1.436; *p* = 0.011). On the other hand, males exhibited a lower risk as age increases (OR = 0.76; 95% CI 0.615–0.928; *p* = 0.008). In conclusion, this study identified that males, multinodular presentation, nodule size, and age are risk factors for the occurrence of splenic neoplasia in dogs, knowledge that will contribute to the diagnostic management of dogs with spleen lesions.

## 1. Introduction

In dogs, splenic nodules can be an incidental finding in ultrasound, radiographic, and surgical examinations [1], or they can be palpable or associated with nonspecific clinical signs [2]. The most frequent clinical signs include weakness, lethargy, anorexia, abdominal distention, and weight loss [1,3]. Patients may present with hemoabdomen and anemia due to ruptured nodules [1], being more frequently associated with splenic malignancies, specifically hemangiosarcoma [4,5]. Splenic lesions can manifest as splenomegaly, which can be diffuse or mainly nodular [6,7]. Cases of small nodules usually do not induce clinical signs [1]. However, independently of their origin, the nodules are grossly similar, and their differentiation is very difficult [1,8]. Several studies have evaluated the diagnostic value of the size of splenic masses using different imaging techniques [9,10,11], although few studies associate the nodule size with histological diagnosis. One of them showed that tumor size could differentiate between hemangiosarcoma and benign lesions [12]; however, the ultrasonographic appearance of these lesions shows poor diagnostic utility for predicting the presence of hemangiosarcoma [13]. Regarding the number of nodules, other ultrasonographic analyses demonstrated a significant association between the presence of multiple nodules with malignant lesions and single nodules with benign lesions [14]. In concordance, it has been described that multiple-nodule hemangiosarcoma has a worse prognosis than focal hemangiosarcoma [15]. Recently, a multivariable model was developed that allows for estimating the probability of splenic malignancy according to a series of presurgical variables, such as the serum protein concentration; the size of the splenic mass; the number of nodules in the spleen, liver, and peritoneum; omentum or mesentery; abdominal effusion; and splenic mass heterogeneity [16]. Nevertheless, this model would have less impact in cases of splenic nodules that are not associated with clinical signs and detected as a finding. In general, there is little information that relates the histopathological diagnosis to the tumor size and the number of nodules in spleen biopsies [12,17]. Some of these have small sample sizes [12] or do not consider all splenic lesions [18]. Thus, this study aimed to characterize a series of splenic lesion cases in dogs, studying potential risk factors for splenic malignancy.

## 2. Methods

This retrospective study was carried out with histopathological reports of splenic nodules in dogs from the archive of an Animal Pathology laboratory of the Metropolitan Region, Chile, between August 2014 and November 2021. Histological diagnosis, breed, sex, age, number of splenic nodules, and size of nodules were recorded. Histological diagnosis was carried out by an American College of Veterinary Pathology (ACVP)-certified Pathologist (Federico F. Cifuentes), according to his experience and morphological criteria, with hematoxylin-eosin staining. Age was categorized as juvenile (up to 12 months old), adult (1–8 years), and senior (equal to or greater than 8 years), as described by McGuire and Gough, 2017 [19]. The number of splenic nodules was determined in the macroscopic description or the previous clinical history and classified as a lesion without nodules, single nodule (one nodule), and multiple nodules (more than one nodule), as previously described [15,20]. The sizes of nodules were defined by the length of the main axis in centimeters (cm) of each nodule that was recorded in the macroscopic description made at the time of sampling the spleen or in the medical history. For multiple nodules, only the largest diameter nodule was considered. The nodule sizes were categorized according to their quartiles into 4 categories (Category 1 ranged from 0.3 to 2 cm; Category 2 from 2.1 to 4 cm; Category 3 from 4.1 to 7 cm; Category 4 from 7.1 to 30 cm).

### Data Analysis

Descriptive statistics tools were used. Thus, the absolute and relative frequencies of each variable were obtained and represented graphically. To evaluate the distribution of the data, the D’Agostino–Pearson test was performed. To compare the size of neoplastic and non-neoplastic nodular lesions, Mann–Whitney test was used. Moreover, a Kruskal–Wallis test was performed to compare nodule sizes between the histological diagnosis of hematoma, hemangiosarcoma, and hyperplasia. A multivariable logistic regression analysis was carried out with a subset of the original dataset (*n* = 285), as the remainder lacked one or more data points, to identify potential risk factors for the presence of splenic neoplasia in dogs with splenic lesions, where the association measure used was an odds ratio (OR) [21,22]. The variables sex, number of nodules, nodule size, age, and sex–age interaction were directly included in the multiple logistic regression model, where the response variable was the presence/absence of splenic neoplasia. Goodness of fit was assessed using the Hosmer and Lemeshow test [23]. A statistical program, R 3.6.3 (R Foundation for Statistical Computing, Vienna, Austria), and GraphPad Prism version 8.0.1 were used. A *p* < 0.05 was considered significant.

## 3. Results

### 3.1. Characterization of the Histopathological Diagnosis of Splenic Lesions in Dogs

Based on the established inclusion criteria (splenic histopathological diagnoses of dogs), 507 reports were analyzed, of which 342 (67.5%) corresponded to non-neoplastic lesions and 165 (32.5%) to neoplastic lesions. The histopathological diagnoses are shown in Table 1. Among the histopathological diagnoses identified, the most frequent was hyperplasia (209 cases, 41.2%), which was subdivided into lymphoid hyperplasia (112 cases, 54%), splenic hyperplasia (77 cases, 37%), histiocytic hyperplasia (13 cases, 6%), and complex hyperplasia (7 cases, 3%). The main neoplastic diagnosis corresponded to hemangiosarcoma, where 99 cases were presented, corresponding to 19.5%. Few secondary lesions (metastasis) were observed.

### 3.2. Characterization of Histological Diagnosis of Splenic Lesions in Dogs According to the Patient’s Parameters

Of the 507 reports analyzed, 342 cases (67.5%) corresponded to purebred dogs, 152 cases (30%) occurred in animals of a mixed breed, and in 13 cases (2.6%), the breed was not recorded in the relevant report (Table 2). Of the pure breeds, the breeds in which splenic lesions were most frequently observed were Cocker Spaniel (50 cases; 14.6%), German Shepherd (50 cases; 14.6%), Labrador Retriever (38 cases; 11.1%), Boxer (24 cases; 7%), Poodle (24 cases; 7%), Golden Retriever (21 cases; 6.1%), Pit Bull Terrier (12 cases; 3.5%), Beagle (11 cases; 3.2%), Fox Terrier (11 cases, 3.2%), Dachshund (9 cases; 2.6%), Rottweiler (7 cases, 2%), Bernese Mountain Dog (6 cases; 1.8%), and others with less than 6 cases each (79 cases; 23.1%). Whether for neoplastic lesions or not, the majority of cases (about 2/3) occurred in individuals of defined breeds. Regarding sex, 231 cases (45.6%) occurred in females, 265 cases (52.3%) in males, and in 11 cases (2.2%), the sex was not recorded in the report, a relation that was maintained in both studied groups. Regarding age, the range of presentation was from 1 to 18 years, with a median in both groups of 10 years. In addition, it was determined that the age range that most frequently presented these lesions was senior dogs (421 cases; 83%). In the adult age range, there were more cases of benign lesions than tumors (Table 2).

### 3.3. Characterization of the Histopathological Diagnoses of Splenic Lesions in Dogs According to Their Macroscopic Characteristics

The macroscopic characteristics evaluated in this section included the size and number of nodules. Splenic lesions were classified according to the number of nodules, where 55 cases (10.8%) did not present any nodule (diffuse splenomegaly), 287 cases (56.6%) presented a single nodule, and 115 cases (22.7%) had two or more nodules. 

Of the total cases, 50 did not record the number of nodules in the report. Regarding the size of the nodules, it is relevant to note that in 200 cases (39.4%), their size was not detailed. The splenic nodule size range was 0.3 to 30 cm, with a median of 4 cm. In the case of non-neoplastic lesions, the size range of the nodules was 0.3–30 cm, with a median of 3.5. In contrast, the size of the nodules of the neoplastic lesions ranged from 1 to 30 cm with a median of 5. Thus, the neoplastic nodules were significantly larger than the non-neoplastic ones (*p* = 0.0016). However, when analyzing the nodular sizes of the most frequent lesions, the sizes of the hematoma nodules reached a median of 6 cm (2–30 cm), significantly larger than that of hemangiosarcoma (median of 5 with ranges of 1.5 and 17 cm) and hyperplasia (median of 3 with ranges of 0.3 and 15 cm) (*p* < 0.0001) (Figure 1).

Of the 507 total cases, only 285 cases were included for logistic regression analysis, as the remainder lacked one or more data points. The nodule sizes were categorized into Category 1, ranging from 0.3 to 2 cm; Category 2 from 2.1 to 4 cm; Category 3 from 4.1 to 7 cm; and Category 4 from 7.1 to 30 cm. To estimate possible risk factors, the variables breed, sex, number of nodules, age, nodule size, and sex–age interaction were incorporated into the multivariable logistic regression model. Thus, only the variables sex, number of nodules, age, nodule size, and sex–age interaction were significant (*p* < 0.05). Regarding sex, males showed 15 times higher odds of having a splenic neoplasia diagnosis than females (OR = 16.21; 95% CI 1.74–150.88; *p* = 0.014). Dogs with two or more splenic nodules had 2.9 times higher odds of having a splenic neoplasia diagnosis than those with a single nodule (OR = 3.94; 95% CI 2.17–7.18; *p* < 0.001). Regarding the size of the nodules, Category 2 had 1.2 times the risk (OR = 2.26; 95% CI 1.04–4.94; *p* = 0.041), Category 3 1.9 times higher odds (OR = 2.96; 95% CI 1.33–6.58; *p* = 0.008), and Category 4 2.6 times higher odds (OR = 3.64; 95% CI 1.56–8.50; *p* = 0.003) of having a diagnosis of splenic neoplasia than Category 1. That is, nodule sizes over 2 cm in diameter had more than twice the odds of having a splenic neoplasia diagnosis than sizes smaller than 2 cm. Increasing age by one unit increased the odds of splenic neoplasia 0.2 times (OR = 1.23; 95% CI 1.05–1.44; *p* = 0.011). Meanwhile, in males, the odds decreased with increasing age (OR = 0.76; 95% CI 0.62–0.93; *p* = 0.008) (Table 3). The model presented a good fit of the data, evidenced by the Hosmer and Lemeshow test (*p* = 0.26).

## 4. Discussion

The spleen is an organ that often develops nodular lesions, especially in older dogs [8]. These lesions can be of benign or malignant origin, and sometimes, there is no availability of efficient diagnostic tools that allow for inferring their origin and optimal therapeutic management [10,11].

In this study, 507 histopathological reports of dogs with splenic lesions were analyzed, in which 74% of the cases corresponded to spleen fragments and 26% to a complete spleen. This result is not under what is recommended by the current literature, where it is suggested to send the complete spleen to analyze at least five different histopathological sections and thus maximize the probability of detection of splenic neoplasia, especially in the context of a diagnosis of hemangiosarcoma [24]. Hemangiosarcoma is the most common malignancy located in the spleen and has a poor prognosis due to the high risk of developing metastasis [5]. It originates from pluripotent bone marrow cells in a pre-differentiation state that migrate to highly vascularized sites, where they undergo malignant transformation. This could explain its high incidence in the spleen [5].

On the other hand, 67.5% of the total cases corresponded to non-neoplastic lesions, results that were consistent with those previously described [25], where a high proportion of non-neoplastic cases were observed. However, several investigations have described that the most frequent splenic lesions are those of a neoplastic origin [12,17,26]. Here, the most frequent neoplastic and non-neoplastic lesions were hemangiosarcoma and hyperplasia, respectively, which coincides with what has been described by other authors [11,15,16,27]. Nevertheless, it has also been described that the most common non-neoplastic lesion is hematoma [17]. Nodular hyperplasia is commonly found in conjunction with splenic hematomas [15,27,28], because it could disrupt blood flow to the marginal sinuses and lead to the pooling of blood in the parenchyma [27]. It has been described that 27% of the cases with hematoma presented concomitant hyperplasia [28], similar results to the present study, where 37% of the cases with hematoma presented some type of hyperplasia. The low proportion of splenic metastases that was observed coincides with what has been described by other authors [1,8], who have reported between 1 and 6% of metastatic splenic lesions in dogs. Within this group, lymphomas in clinical stage IV, histiocytic tumors, mast cell tumors, multiple myelomas, and different types of leukemia are included, as well as various types of sarcomas, carcinomas, and neuroendocrine tumors. These outcomes could be explained by the presence of macrophages in the sinuses and splenic cords that can recognize and destroy embolized cancer cells and prevent their subsequent colonization [20].

On the other hand, it has been described in humans that the low frequency of splenic metastases could be associated with a high concentration of lymphoid tissue and with mechanical factors such as the contractile nature of the spleen and its constant blood flow, which would prevent the implantation of metastatic cancer cells in the organ [29,30]. In addition, the absence of afferent lymphatics limits lymphogenic metastasis [29,30]. In the spleen of animals, metastatic sarcoma is more frequent than carcinoma [31], which differs from this study, where the majority were metastatic carcinomas.

The breeds that were more affected with splenic lesions are those with weights greater than 25 kg [5], such as German Shepherd, Golden Retriever, and Labrador Retriever [5,10,15,17,32], which coincides with what was observed in this study, although many cases were also observed in Cocker Spaniels. These data suggest a genetic susceptibility; however, to date, it has not been mechanistically elucidated why these breeds have a greater risk of developing splenic lesions.

Regarding age, the most affected group was senior dogs (83%), with a median age of 10 years, which coincides with what has been described by other authors, who have also reported a median age of presentation at 10 years [33]. In concordance, this variable is a risk factor for the presentation of splenic malignancy. Aging is associated with a lower immunological competence, less efficient DNA repair, a greater number of damaged tumor suppressor genes, and a lower number and function of mitochondria. In addition, the functional reserves of the organic systems in geriatric animals are decreased, in addition to sarcopenia and an increase in the prevalence of chronic diseases, which can cause fragility and stress, implying a greater susceptibility to developing cancer [34].

No predilection by sex has been described in the literature for the different splenic lesions in dogs [8]. However, some studies have shown a higher proportion of affected males than females [1,5,27,33], which coincides with the results of the present study, in which males showed an increased risk of developing splenic neoplasia. In this regard, it has been described that late-castrated females have a higher risk of developing tumors such as hemangiosarcoma and mast cell tumors than reproductively intact individuals [35,36], which suggests a possible protective role of estradiol [35]. Additionally, spayed females have exhibited a higher risk of developing splenic hemangiosarcoma than intact females [32]. Luteinizing hormone (LH) stimulates the synthesis and secretion of estradiol and progesterone from the ovaries, and these steroids exert negative feedback on LH synthesis. In cases of gonadectomized female dogs, the absence of steroids induces a loss of negative feedback; therefore, LH concentrations rise significantly. In this context, the expression of LH receptors has been described in different cells, including vascular endothelial cells, vascular smooth muscle cells, and hemangiosarcoma cells. In this way, LH could act as a pro-carcinogenic agent through the induction of cell proliferation and nitric oxide synthesis. The above-described factors could explain why spayed female dogs show a greater risk of developing hemangiosarcoma, a hypothesis that needs to be tested [37].

Various studies have reported that splenic nodular lesions are more frequent than diffuse splenomegaly [1,15]. In this context, lesions with a single nodule are the most prevalent [15], which coincides with the results of the present study. It is important to consider that this result could be biased, given that normally, spleens with a single nodule are most likely to be further analyzed. This feature would have prognostic value, since patients with hemangiosarcoma with a single nodule exhibit better survival time than those specimens with multinodular hemangiosarcoma [15]. Moreover, the presence of a single nodule has been associated with a benign lesion, while multiple nodule lesions with a similar ultrasonographic appearance are associated with a malignant lesion [14]. This is coherent with what is described in this report, where lesions with multiple nodules were mostly associated with splenic neoplasia.

In this study, it was observed that neoplastic lesions were significantly larger than non-neoplastic ones, which is consistent with previous studies showing that the size of malignant nodules is larger than that of benign ones, demonstrated by using ultrasound [11]. Additionally, it has been observed that 95% of malignant splenic tumors had diameters that were greater than 2.5 cm, and 95% of benign tumors had diameters that were less than 2.5 cm on ultrasound [11], which is consistent with this study, where nodular diameters that were greater than 2 cm increased the probability that the morphological diagnosis was neoplastic. In this regard, it has recently been described that computed tomography can be used to differentiate focal splenic lesions. Sarcomas are usually large, have a cystic appearance, and have low post-contrast enhancement. On the other hand, benign lesions are small, solid, and have high post-contrast enhancement [38].

When evaluating the size of the main splenic lesions, the present study revealed that the hematoma is significantly larger than the hemangiosarcoma, which is consistent with other studies, where it is stated that it is not possible to distinguish between a hemangiosarcoma and a splenic hematoma by size or other gross findings [27,33]. Furthermore, it is also not possible to differentiate between a hemangiosarcoma and a hematoma by ultrasound, as both lesions exhibit extensive hemorrhage and necrosis [39]. However, computed tomography and elastography are advanced diagnostic techniques that can differentiate between malignant and benign splenic lesions [40,41]. In this context, it has been reported that a shear wave velocity (SWV)—a parameter obtained by elastography—of >2.6 m/s indicates malignancy in splenic lesions with a sensitivity of 95% and specificity of 100% [41]. Acoustic radiation force impulse (ARFI) elastography is an ultrasound-based technique that allows for the differentiation of malignant neoplastic lesions in the thyroid gland, mammary gland, and lymph nodes, among others, through a quantitative evaluation that includes the measurement of the SWV. This allows for the evaluation of the rigidity of the tumor lesions, which is high in case of a malignancy due to a higher collagen III content, fibrosis, and areas of microcalcification; however, its results have not been associated with specific histopathological diagnoses. Thus, this technique is a non-invasive, objective, and operator-independent tool that can improve the diagnostic approach in case of nodular lesions in the spleen [41].

Although splenectomy is well tolerated in dogs, and recovery times after surgery are short [10], it can have some consequences for the patient, such as exercise intolerance, decreased response to hypoxia, and susceptibility to hemoparasites [11]. Therefore, it is important to consider the epidemiological data exposed here when deciding on a therapeutic procedure like this.

The limitations of this study involved data loss that reduced the sample size in the logistic analysis. Another limitation is that data on the patient’s reproductive status were not recorded, which did not allow for analysis of the potential association of this condition with the risk of splenic malignancy, a relation that is described above [35,36]. The results related to the greater presence of single nodules could be biased, given that normally, spleens with a single nodule are most likely to be further analyzed. Finally, considering clinical data would have strengthened this analysis.

## 5. Conclusions

The most frequent neoplastic and non-neoplastic histopathological diagnoses recorded in this study were hemangiosarcoma and hyperplasia, respectively. Splenic lesions of a neoplastic nature are larger than non-neoplastic lesions; however, hematomas have a larger median size than hemangiosarcoma. Most of the splenic lesions described occurred in senile and purebred individuals such as Cocker Spaniel, German Shepherd, and Labrador Retriever. The risk factors for the presence of splenic neoplasia that were detected in this study were sex (male), the presence of multiple nodules, and increasing age and nodular size. Dogs with nodular splenic lesions that were larger than 2 cm showed a higher risk of having a diagnosis of splenic malignancy.

## Figures and Tables

**Figure 1 animals-14-00802-f001:**
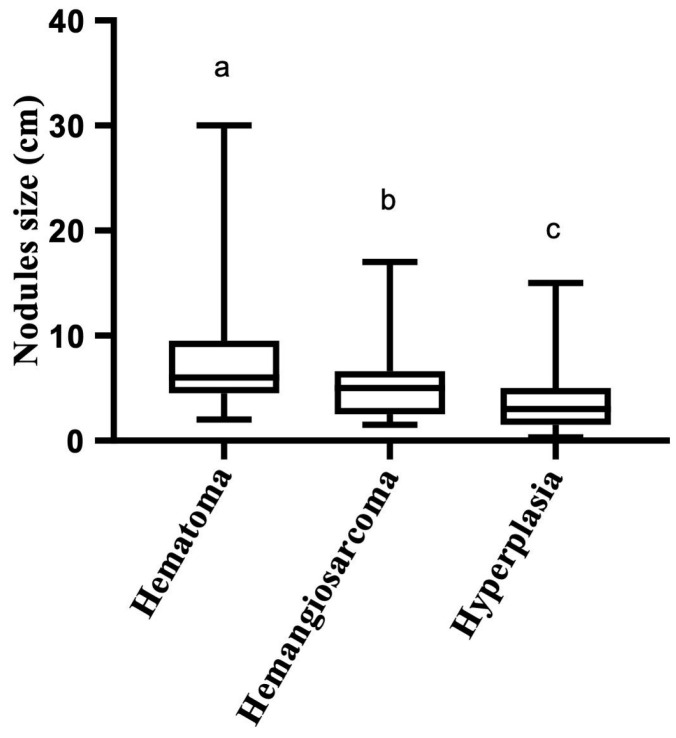
Splenic nodule sizes according to morphological diagnosis (hematoma, hemangiosarcoma, and hyperplasia). Hematomas were larger than hemangiosarcoma and hyperplasia. Hemangiosarcoma were larger than hyperplasia. Different letters denote significant differences. *p* < 0.0001. Kruskal–Wallis test.

**Table 1 animals-14-00802-t001:** Distribution of non-neoplastic and neoplastic splenic lesions according to histopathological diagnosis.

Histopathological Diagnosis	*n*	%
Non-neoplastic	342	67.5
Hyperplasia	209	41.2
Hematoma	84	16.6
Congestion	18	3.6
Splenitis	10	2.0
Fibrohistiocytic nodules	8	1.6
Other lesions	13	2.6
Neoplastic	165	32.5
Hemangiosarcoma	99	19.5
Undifferentiated sarcoma	23	4.5
Lymphoma	15	3.0
Myelolipoma	7	1.4
Histiocytic sarcoma	4	0.8
Fibrosarcoma	3	0.6
Hemangioma	2	0.4
Leiomyosarcoma	2	0.4
Lipoma	2	0.4
Plasma cell tumor	2	0.4
Metastasis	6	1.2
Total	507	100

**Table 2 animals-14-00802-t002:** Distribution of canine splenic lesions according to breed, sex, and age.

Variables	Splenic Injuries *n* (%)	Neoplastic *n* (%)	Non-Neoplastic *n* (%)
Breed			
Pure	342 (67.4)	111 (67.3)	231 (67.5)
Mixed	152 (30)	50 (30.3)	102 (29.8)
Total	507 (100)	165 (100)	342 (100)
Sex			
Female	231 (45.6)	76 (46.1)	155 (45.3)
Male	265 (52.3)	86 (52.1)	179 (52.3)
Not indicated	11 (2.2)	3 (1.8)	8 (2.4)
Total	507 (100)	165 (100)	342 (100)
Age			
Juvenile	2 (0.4)	0	2 (0.6)
Adult	60 (11.8)	14 (8.5)	46 (13.5)
Senior	421 (83)	141 (85.5)	280 (81.9)
Not indicated	24 (4.8)	10 (6)	14 (4)
Total	507 (100)	165 (100)	342 (100)

**Table 3 animals-14-00802-t003:** Multiple logistic regression for the presence of splenic neoplasia in dogs.

Variables	Categories	OR	95% CI		*p*-Value
			Lower	Upper	
(Intercept)		0.018	0.003	0.116	<0.001
Sex	Male	16.205	1.741	150.879	0.014
	Female		Reference		
Nodules number	Single		Reference		
	Multiple	3.944	2.168	7.177	<0.001
Nodules size	0.3–2 cm		Reference		
	2.1–4 cm	2.262	1.036	4.941	0.041
	4.1–7 cm	2.958	1.331	6.576	0.008
	7.1–30 cm	3.644	1.562	8.499	0.003
Age		1.227	1.048	1.436	0.011
Sex–age interaction	Male	0.756	0.615	0.928	0.008
	Female		Reference		

## Data Availability

All the data are published in this manuscript or can be obtained upon reasonable request.

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
