# Peer review of "Pathological Characterization and Risk Factors of Splenic Nodular Lesions in Dogs (Canis lupus familiaris)"

_animals, 2024, doi:10.3390/ani14050802_

Round 1

Reviewer 1 Report

Comments and Suggestions for Authors

Dear Author, 

In my opinion, this well-written, orderly, manuscript is interesting and contains organized knowledge. You describe the retrospective analysis of splenic lesions divided into neoplastic and non-neoplastic lesions. I found no errors or omissions. The list of references is long, from 1988 to 2023.

 I think, would be a very interesting comparison of elastography with histopathological findings in the same cases. It's a pity that it doesn't have this.  You did not provide these studies and only cited other scientists. 

Best regards

SD

Author Response

We appreciate the reviewer's comments. A paragraph has been added (lines 295-303) that explains the usefulness of elastography in the definition of malignancy in cases of nodular lesions of the spleen.

Reviewer 2 Report

Comments and Suggestions for Authors

This retrospective study of 507 histological reports was aiming at identifying risk factors for splenic nodular lesions to be a) neoplastic, and/or b) malign.

Male, purebred and progressed age are risk factors for neoplastic lesions. Lesions larger than 2 cm were more likely to be neoplastic.

I believe that your data are valuable for the community but - to my mind – biased. A major review is necessary to solve major concerns. I hope you consider the following lines helpful for improving your manuscript.

Major concerns

The title does not fit to the content. In my opinion did not do a risk analysis, you looked for macro- (n=507-200?) and microscopic (n= 507) findings associated with neoplastic or non-neoplastic splenic disease; hence *analysis* in the title should be replaced with *factors* OR restate as *Retrospective analysis of histologic reports in splenic samples* which is true in 507 cases. Clarify at all instances the exact number of cases you could analyze for your research question (which you might want to modify, e.g. did you look at masses or nodules?). Reason: Nodules were not present in 55 cases. Full spleen was only available in 307 cases. You might even want to remove cases to improve the inclusion criteria and redo the statistics.

Are you sure that you can exclude a biased sample? Spleens with single nodules are most likely to be further analyzed. Potential sample bias (very likely) should be added to limitations of this study; currently I do not agree with the analysis in lines 234 and 235.

Pure breeds frequently associated with splenic lesions are not listed in the abstract. Why? It might be a risk factor to be a German Shephard Dog, especially because they tend to have large spleens. You did not comment on dog pure breeds (apart from pure- or non-pure breed).

I find the terminology regarding age vague: Senior is classified in Material and Methods but in the text, you use the word senile (-> same as senior?); please review expressions for age and clarify the meanings. This appears to be important because multiple logistic regression showed that risk for male dogs (L40 which risk?) changes with progressed age.

Minor problems

I cannot explain the case number (n= 507) in chapter 3.2, e.g. you list 342 pure breed dogs, but the number of pure breeds listed adds up to 242. My fault?
Single nodes might be more likely to be analyzed than multiple (when being a clinician); a single benign nodule can be an early stage of malignant disease too. Please comment on this.

Comments on the Quality of English Language

Review for typos, here are some I identified.

L114: Complex -> complex

L118: , leiomyosarcoma (not dot)

L119: add: and plasma cell tumor…

Figure 1. Review small or large letters for names of diseases.

Table 3: 0.3 (not 0,3)

L261: replace ->with words

Table 1: no-neoplastic versus non-neoplastic; please review / harmonize these expressions

Table 1: Hiperplasia (-> Hyperplasia); Mielo (-> Myelo)

Author Response

Response to comment 1 (R1): We very much appreciate the comments. The title was modified according to the reviewer's comment, "risk analysis" was replaced with "risk factors". In terms of the statistical analysis, the logistic multivariable regression model was performed with a subset of the original dataset, due to a lack of full details on factors to evaluate in the model. Those individuals who lacked information, for example on the size of the nodules, were excluded from the analysis, leading to a clean dataset. Further details have been added to the Data analysis subsection.

R2: We very much appreciate the comments. Limitation was added at lines 264 - 267. As stated in the Methods section, samples correspond to the archive of an Animal Pathology laboratory, run by a certified Pathologist, this is important given that in Chile this is not a massive situation, meaning that few Universities and private laboratories perform histopathology as a service, those receiving the majority of the samples, also there are few certified Veterinary Pathologists in Chile, most of them concentrated in the Metropolitana Region (< than 5), this is important to state because all the histopathology studies are concentrated in the same services, with this even when there could be a sample bias, this services concentrate the majority of the samples in the region and potentially in Chile, this study concentrated the cases of around seven years considering only splenic nodular lesions.

R3: The most observed races were included in the abstract and the discussion (lines 231-236)

R4: Thank you very much for the comment. The necessary adjustments were made. The concept "senior" referring to dogs over 8 years of age is mentioned throughout the text.

R5: A pertinent comment was added in the results section (lines 136-137), as the reviewer is correct in his comment. Thank you so much. From a clinical point of view, always in the case of the presence of single or multiple splenic nodules, at least an ultrasound follow-up is performed to see how they evolve. Indeed, single benign-looking nodules can be malignant at an early stage.

Regarding the minor comments, we appreciate these observations which were corrected

Reviewer 3 Report

Comments and Suggestions for Authors

This is an interesting study on histological classification and risk factors for splenic neoplasia, a common condition in veterinary practice. Please find below some comments and suggestions to improve the manuscript. There are some sections in which English is hard to understand; however, the general text is well-written and easy to catch. The online program Grammarly would be a helpful tool to help fix this issue.

Line 20 – It was determined...

Line 29 – A total of 507...

Lines 35-38 – Please include the respective 95% confidence intervals for each OR presented. When this kind of analysis is performed, the p-value can misrepresent results as significant. However, analysis of the 95%CI better represents statistical significance when the interval does not cross the trivial value 1).

Line 42 – Ideally, keywords should not repeat words in the title. This detail helps the paper to be found easily in database searches.

Line 51 – please do not capitalize Hemangiosarcoma

Introduction – general comments - This section needs to be carefully reviewed regarding English grammar to make it easier to understand. Some sentences seemed to lose sense (e.g., line 53)

MMs

Line 80 – can the authors provide more details regarding the criteria used for lesion differentiation? Was any IHC applied, or did the classification system adopted rely on HE characteristics and pathologist expertise?

Data analysis – general comments – more detail should be given regarding neoplastic and non-neoplastic lesions comparison by the Mann-Whitney test. What exactly was being compared (number of lesions, size of lesions, age of patients, etc…)? Please, consider including 95%CI for each OR reported. This is more important them the p-value (that can be kept).

Line 114 - please do not capitalize Complex

Table 1 can completely replace the preceding paragraph since the same information is being shown in the text and table. The text could just mention that histopathological diagnoses are shown in Table 1, emphasizing the greater occurrence of non-neoplastic lesions or other general aspects the authors would like to give emphasis. The table is fine and easier to understand the distribution of the results.

Line 125 - … (67,5%) correspond to pure breed dogs, 152 cases...

Table 2 - please move the table to before item 3.3. Also, I suggest removing “injuries” from the second and third columns, leaving just neoplastic and non-neoplastic.

Lines 170-180 – Please, to deliver complete information, please report the p-values after each 95%CI [e.g., (OR 3.94; 95% CI 2.17-7.18; p = x.xxxx). Another critical issue is that when OR analysis is employed in retrospective studies, what is being studied is the chance of a given factor being more associated with a given outcome. In this case, an OR of 3 represents a 2-fold higher chance, not 3, since the trivial value 1 is null. Please correct this paragraph considering this concept

Line 187 - please do not capitalize Hemangiosarcoma

Table 3 – please increase the space in the first collum, and reduce the space in the second, so the general aspect turns more linear. Also, please move this table up, out of the discussion section.

Line 230 – despite the eventual protective effect of estradiol in intact females, one cannot ignore that eventually increased tumor rates in neutered/spayed individuals can be secondary to chronically increased LH concentrations after neutering. A mention of this hypothesis could be included. More can be read in this paper by Kunzler 2020 (doi: 10.3390/ani10040599).

Comments on the Quality of English Language

In general is well written, however, some sections are harder and can be improved. 

Author Response

We appreciate all the comments made by the reviewer. We detail the answers below:

Abstract: we correct the observations in the abstract, including the description of the confidence intervals obtained, and keyword modification

Introduction: we improve English grammar

MMs: We do not use IHC to differentiate some lesions. Dr. Cifuentes is an ACVP Diplomate and has considerable experience in the microscopic evaluation of spleen lesions. We know that in the case of some neoplastic lesions, the ideal would be to confirm with IHC, however, it seems to us that the morphological diagnosis made here allows a study like this to be carried out. Between lines 83-86, these details are described.

Results: We consider and appreciate all comments made, including the correction of Tables 2 and 3.

Discussion: We consider the information suggested by the reviewer regarding the eventual LH role in hemangiosarcoma incidence (lines 254-262).

Round 2

Reviewer 2 Report

Comments and Suggestions for Authors

Well done. To me, there are no concerns left. A final language editing might be necessary.

e.g.
L72 inhomogeneity -> heterogeneity (like black and white and NOT inblack)
L117 histopathological diagnosis (not plural, since each animal has one, hopefully) is shown..

and I suggest to add sums / total values in table 2, like you did in table 1.

Comments on the Quality of English Language

Well done. To me, there are no concerns left. A final language editing might be necessary.

e.g.
L72 inhomogeneity -> heterogeneity (like black and white and NOT inblack)
L117 histopathological diagnosis (not plural, since each animal has one, hopefully) is shown..

Author Response

We appreciate the comments and have incorporated the new suggestions into the manuscript 

Reviewer 3 Report

Comments and Suggestions for Authors

Thank you for considering the suggested modifications. Nice work!

Author Response

We appreciate the comments